# Pro-Inflammatory Chemokines CCL5, CXCL12, and CX3CL1 Bind to and Activate Platelet Integrin αIIbβ3 in an Allosteric Manner

**DOI:** 10.3390/cells11193059

**Published:** 2022-09-29

**Authors:** Yoko K. Takada, Masaaki Fujita, Yoshikazu Takada

**Affiliations:** 1Department of Dermatology, School of Medicine, University of California–Davis, 4645 Second Ave., Research III Suite 3300, Sacramento, CA 95817, USA; 2Department of Biochemistry and Molecular Medicine, University of California Davis School of Medicine, Sacramento, CA 95817, USA

**Keywords:** integrin activation, chemokine, platelet

## Abstract

Activation of platelet integrin αIIbβ3, a key event for hemostasis and thrombus formation, is known to be mediated exclusively by inside-out signaling. We showed that inflammatory chemokines CX3CL1 and CXCL12 in previous studies, and CCL5 in this study, bound to the allosteric binding site (site 2) of vascular integrin αvβ3, in addition to the classical ligand binding site (site 1), and allosterically activated integrins independent of inside-out signaling. Since αIIbβ3 is exposed to inflammatory chemokines at increased concentrations during inflammation (e.g., cytokine/chemokine storm) and platelet activation, we hypothesized that these chemokines bind to and activate αIIbβ3 in an allosteric activation mechanism. We found that these chemokines bound to αIIbβ3. Notably, they activated soluble αIIbβ3 in 1 mM Ca^2+^ by binding to site 2. They activated cell-surface αIIbβ3 on CHO cells, which lack machinery for inside-out signaling or chemokine receptors, quickly (<1 min) and at low concentrations (1–10 ng/mL) compared to activation of soluble αIIbβ3, probably because chemokines bind to cell surface proteoglycans. Furthermore, activation of αIIbβ3 by the chemokines was several times more potent than 1 mM Mn^2+^. We propose that CCL5 and CXCL12 (stored in platelet granules) may allosterically activate αIIbβ3 upon platelet activation and trigger platelet aggregation. Transmembrane CX3CL1 on activated endothelial cells may mediate platelet–endothelial interaction by binding to and activating αIIbβ3. Additionally, these chemokines in circulation over-produced during inflammation may trigger αIIbβ3 activation, which is a possible missing link between inflammation and thrombosis.

## 1. Introduction

Integrins are a superfamily of cell-surface heterodimers that recognize extracellular matrix (ECM) proteins, cell-surface ligands (e.g., VCAM-1 and ICAM-1), and soluble ligands (e.g., growth factors such as FGF and IGF) [1,2,3]. Integrins play important roles in normal biological process (e.g., wound healing and hemostasis) and in the pathogenesis of diseases [4]. By virtual screening of the protein data bank using docking simulation, we identified the chemokine domain of CX3CL1 [5] as a new ligand for integrins αvβ3, α4β1, and α5β1, and that CX3CL1 and integrins simultaneously bound to CX3CR1 [5]. The CX3CL1 mutant defective in integrin binding was defective in signaling and acted as an antagonist, although it still bound to CX3CR1 [5], indicating that the direct integrin binding to CX3CL1 and subsequent integrin-CX3CL1-CX3CR1 ternary complex formation is required for its signaling functions. CX3CL1 is expressed on the cell surface of IL-1β- and TNFα-activated endothelium as a membrane-bound form [6]. Soluble CX3CL1 is released by metalloproteinases ADAM10 and ADAM17 [7,8,9]. Interaction between membrane-bound CX3CL1 and CX3CR1 promotes leukocyte adhesion to the endothelium [10,11,12].

We unexpectedly discovered that CX3CL1 activated soluble integrin αvβ3 in 1 mM Ca^2+^ in cell-free conditions by binding to the second ligand-binding site (site 2), which is distinct from the classical ligand-binding site (site 1) in the integrin headpiece [13]. We showed that peptides from site 2 of the integrin β subunit bound to CX3CL1 and blocked integrin activation by CX3CL1 [13]. Therefore, we concluded that site 2 is the allosteric binding site and involved in allosteric activation of integrins. We showed that CXCL12 activated integrins αvβ3, α4β1, and α5β1 by binding to site 2 [14], indicating that this mechanism of integrin activation is not limited to CX3CL1. CXCL12 is a potent chemoattractant for leukocytes and is believed to regulate signaling events through CXCR4 and CXCR7 in leukocytes [15,16,17,18]. CXCL12 (Stromal cell-derived factor 1, SDF-1) is ubiquitously expressed in many tissues and cell types. Binding of CXCL12 to CXCR4 induces heterotrimeric G protein signaling leading to activation of the pro-inflammatory signaling [19,20]. CXCL12/CXCR4 are over-expressed in rheumatoid arthritis, systemic lupus erythematosus, multiple sclerosis, and inflammatory bowel disease [21,22,23,24]. CXCL12 is stored in platelet granules and rapidly transported to the surface upon platelet activation [25].

Platelet integrin αIIbβ3 is a receptor for several proteins including fibronectin, fibrinogen, plasminogen, prothrombin, thrombospondin, vitronectin, and von Willebrand factor (VWF) [3]. Activation of αIIbβ3 is a key event that triggers platelet aggregation by inducing αIIbβ3 binding to fibrinogen leading to bridge formation between platelets [26,27]. The mechanism of integrin activation has been extensively studied in integrin αIIbβ3 as a model. Integrin activation is defined by the increase in binding to monomeric ligand and we have used monomeric fragments of fibrinogen or fibronectin.

It has been proposed that activation of integrin αIIbβ3 is induced exclusively by inside-out signaling by platelet agonists (e.g., thrombin, ADP, collagen), which bind to receptors on the cell surface. Signals received by other receptors induce the binding of talin and kindlin to cytoplasmic end of the integrin β subunit at sites of actin polymerization [26,27,28]. However, it has been proposed that canonical integrin activation pathways by platelet agonists induces integrin binding to multivalent ligands (e.g., ligand-mimetic antibody, Pac-1 IgM specific for αIIbβ3, with potential 10-binding sites), but does not enhance ligand binding affinity to monovalent ligand [29,30]. 

We showed that the affinity to monomeric ligands is increased by chemokines CX3CL1 and CXCL12 in non-αIIbβ3 integrins, but it is unclear if this occurs in platelet αIIbβ3. Notably, CXCL12 is stored in platelet granules and rapidly transported to the surface upon platelet activation. We thus hypothesized that other chemokines that are stored in platelet granules (e.g., CCL5) are potentially involved in allosteric activation of platelet integrin αIIbβ3 and trigger thrombus formation. So, we decided to include CCL5 in the present study. It is important to study whether the binding of chemokines CCL5, CX3CL1, and CXCL12 to site 2 activates αIIbβ3, a model system for integrin activation. 

CCL5 (Rantes) is a pro-inflammatory chemokine, recruiting leukocytes to the site of inflammation. CCL5 is chemotactic for T cells, eosinophils, and basophils, monocytes, natural-killer (NK) cells, dendritic cells, and mastocytes [31]. CCL5 is mainly expressed by T-cells and monocytes [32] and it is abundantly expressed by epithelial cells, fibroblasts, and thrombocytes. CCL5 is involved in numerous human diseases and disorders, including transplantation rejection [33], anti-viral immunity [31], tumor development [34], and viral hepatitis or COVID-19 [32]. 

In the present study, we first describe that CCL5 bound to and activated vascular integrin αvβ3, as in CX3CL1 and CXCL12. We describe that CCL5, CXCL12, and CX3CL1 specifically bound to αIIbβ3 to site 1, indicating that they are new ligands for αIIbβ3. These chemokines activated soluble αIIbβ3 by binding to site 2. In addition, cell-surface αIIbβ3 on CHO cells, which lack the machinery for inside-out signaling or cognate receptors for these chemokines, was more efficiently activated by these chemokines probably because they were concentrated by binding to surface proteoglycans. These findings indicate that αIIbβ3 can be activated in an allosteric manner by CXCL12 and CCL5 stored in platelet granules, and CX3CL1, which is expressed on the cell surface (e.g., activated endothelial cells). We propose a new mechanism of αIIbβ3 activation by chemokines in the platelet granules upon platelet activation. Additionally, exogenous chemokines during inflammation (e.g., cytokines storms or autoimmune diseases) might be a possible missing link between inflammation and thrombosis.

## 2. Experimental Procedures

Materials. cDNA encoding (6 His tag and Fibrinogen γ-chain C-terminal residues 390-411) [HHHHHH]NRLTIGEGQQHHLGGAKQAGDV] was conjugated with the C-terminus of GST (designated γC390-411) in pGEXT2 vector (BamHI/EcoRI site). The protein was synthesized in E. coli BL21 and purified using glutathione affinity chromatography. CHO cells that express recombinant human αIIbβ3 were described [35]. The fibrinogen γ-chain C-terminal domain (γC151-411) was generated as previously described [36]. CX3CL1 [5] and CXCL12 [14] were synthesized as described. cDNA encoding mature CCL5 (SPYSSDTTPCCFAYIARPLPRAHIKEYFYTSGKCSNPAVVFVTRKNRQVCANPEKKWVREYINSLEMS) was synthesized and subcloned into the BamH1/EcoR1 site of pET28a vector. Protein synthesis was induced by IPTG in E. coli BL21. Proteins were produced as insoluble inclusion bodies and purified in denaturing conditions and refolded as described [5]. The disintegrin domain of ADAM15 fused to GST (ADMA15 disintegrin) and parent GST were synthesized as previously described [37].

Cyclic β3 site 2 peptide fused to GST-The 29-mer cyclic β3 site 2 peptide C260-RLAGIV[QPNDGSHVGSDNHYSASTTM]C288 was synthesized by inserting oligonucleotides encoding this sequence into the BamHI-EcoRI site of pGEX-2T vector. The positions of Cys residues for disulfide linkage were selected by using Disulfide by Design-2 (DbD2) software (http://cptweb.cpt.wayne.edu/DbD2/) [38]. It predicted that mutating Gly260 and Asp288 to Cys disulfide-linked cyclic site 2 peptide of β3 does not affect the conformation of the original site 2 peptide sequence QPNDGSHVGSDNHYSASTTM in the 3D structure. We found that the cyclic site 2 peptide bound to CX3CL1 and sPLA2-IIA to a similar extent to non-cyclized β3 site 2 peptides in ELISA-type assays (data not shown). 

### 2.1. Binding of Site 2 Peptide to CCL5

Wells of 96-well Immulon 2 microtiter plates (Dynatech Laboratories, Chantilly, VA, USA) were coated with 100 µL PBS containing CCL5 for 2 h at 37 °C. Remaining protein binding sites were blocked by incubating with PBS/0.1% BSA for 30 min at room temperature. After washing with PBS, GST-tagged cyclic site 2 peptide was added to the wells and incubated in PBS for 2 h at room temperature. After unbound GST-tagged site 2 peptides were removed by rinsing the wells with PBS, bound GST-tagged site 2 peptides were measured using HRP-conjugated anti-GST antibody and peroxidase substrates. 

### 2.2. Binding of Soluble αIIbβ3 to Chemokines

ELISA-type binding assays were performed as described previously [5]. Briefly, wells of 96-well Immulon 2 microtiter plates (Dynatech Laboratories, Chantilly, VA, USA) were coated with 100 µL PBS containing chemokines for 2 h at 37 °C. The remaining protein binding sites were blocked by incubating with PBS/0.1% BSA for 30 min at room temperature. After washing with PBS, soluble recombinant αIIbβ3 (AgroBio, 1 µg/mL) was added to the wells and incubated in HEPES-Tyrodes buffer (10 mM HEPES, 150 mM NaCl, 12 mM NaHCO_3_, 0.4 mM NaH_2_PO_4_, 2.5 mM KCl, 0.1% glucose, 0.1% BSA) with 1 mM MnCl_2_ for 1 h at room temperature. After unbound αIIbβ3 was removed by rinsing the wells with binding buffer, bound αIIbβ3 was measured using anti-integrin β3 mAb (AV-10) followed by HRP-conjugated goat anti mouse IgG and peroxidase substrates.

### 2.3. Activation of Soluble αIIbβ3 by Chemokines

ELISA-type binding assays were performed as described previously [14]. Briefly, wells of 96-well Immulon 2 microtiter plates were coated with 100 µL PBS containing γC390-411 for 2 h at 37 °C. Remaining protein binding sites were blocked by incubating with PBS/0.1% BSA for 30 min at room temperature. After washing with PBS, soluble recombinant αIIbβ3 (1 µg/mL) was pre-incubated with chemokines for 10 min at room temperature and was added to the wells and incubated in HEPES-Tyrodes buffer with 1 mM CaCl_2_ for 1 h at room temperature. After unbound αIIbβ3 was removed by rinsing the wells with binding buffer, bound αIIbβ3 was measured using anti-integrin β3 mAb (AV-10) followed by HRP-conjugated goat anti mouse IgG and peroxidase substrates. 

### 2.4. Activation of Cell-Surface αIIbβ3 by Chemokines

αIIbβ3-CHO cells were cultured in DMEM/10% FCS. The cells were resuspended with HEPES-Tyrodes buffer/0.02% BSA (heat-treated at 80 °C for 20 min to remove contaminating cell adhesion molecules). The αIIbβ3-CHO cells were then incubated with chemokines for 30 min on ice and then incubated with FITC-labeled γC390-411 (50 µg/mL) for 30 min at room temperature. The cells were washed with PBS/0.02% BSA and analyzed in BD Accuri flow cytometer (Becton Dickinson, Mountain View, CA, USA) or Attune flow cytometer (ThermoFischer Scientific, Waltham, MA, USA). For time-course studies, FITC-labeled γC390-411 was added to cell suspension first and the mixture was kept on ice, and then incubated with chemokines for the time indicated before being analyzed in flow cytometry. The data were analyzed using FlowJo 7.6.5, BD Bioscience, Franklin Lakes, NJ, USA.

### 2.5. Docking Simulation

Docking simulation of interaction between CCL5 (1EQT.pdb) and integrin αvβ3 (open headpiece form 1L5G, or closed headpiece form, PDB code 1JV2) was performed using AutoDock3 as described previously [1]. We used the headpiece (residues 1–438 of αv and residues 55–432 of β3) of αvβ3. Cations were not present in integrins during docking simulation. The ligand is presently compiled to a maximum size of 1024 atoms. Atomic solvation parameters and fractional volumes were assigned to the protein atoms by using the AddSol utility, and grid maps were calculated by using AutoGrid utility in AutoDock 3.05. A grid map with 127 × 127 × 127 points and a grid point spacing of 0.603 Å included the headpiece of αvβ3. Kollman ‘united-atom’ charges were used. AutoDock 3.05 uses a Lamarckian genetic algorithm (LGA) that couples a typical Darwinian genetic algorithm for global searching with the Solis and Wets algorithm for local searching. The LGA parameters were defined as follows: the initial population of random individuals had a size of 50 individuals; each docking was terminated with a maximum number of 1 × 106 energy evaluations or a maximum number of 27,000 generations, whichever came first; mutation and crossover rates were set at 0.02 and 0.80, respectively. An elitism value of 1 was applied, which ensured that the top-ranked individual in the population always survived into the next generation. A maximum of 300 iterations per local search were used. The probability of performing a local search on an individual was 0.06, whereas the maximum number of consecutive successes or failures before doubling or halving the search step size was 4. 

### 2.6. Statistical Analysis

Treatment differences were tested using ANOVA and a Tukey multiple comparison test to control the global type I error using Prism 7 (GraphPad Software, San Diego, CA, USA).

## 3. Results

### 3.1. CCL5 Binds to and Activates Integrin αvβ3

The primary goal of this study is to determine if three chemokines bind to and activate integrin αIIbβ3. Active and inactive conformations, 1L5G.pdb (open headpiece αvβ3) and 1JV2 (closed-headpiece αvβ3), are defined in αvβ3 but not in αIIbβ3. We have shown that CX3CL1 binds to the classical ligand (RGD)-binding site (site 1) in open headpiece αvβ3 and to the allosteric site (site 2) in a closed headpiece αvβ3 in docking simulation [13]. Additionally, CXCL12 bound to site 2 and activated integrins αvβ3, α4β1, and α5β1 [14]. It is unclear if CCL5, a major chemokine in platelet granules, binds to and activates any integrins. Thus, we first studied whether CCL5 binds to site 1 and site 2 of αvβ3 before we studied CCL5 binding to αIIbβ3. Our docking simulation between integrin αvβ3 and CCL5 predicted that CCL5 binds to site 1 at high affinity (docking energy −24.8 kcal/mol) (Figure 1a) and to site 2 (docking energy −20.2 kcal/mol) (Figure 1b). Amino acid residues predicted to be involved in site 1 and site 2 binding are shown in Table 1 and Table 2. 

We studied whether αvβ3 binds to CCL5 in ELISA-type binding assays. CCL5 was immobilized to wells of 96-well microtiter plates and incubated with soluble αvβ3 in 1 mM Mn^2+^. We found that soluble αvβ3 bound to immobilized CCL5 in a dose-dependent manner (Kd = 6.7 µg/mL) (Figure 2a). The αvβ3 binding to CCL5 required Mn^2+^, and EDTA, Mg^2+^, and Ca^2+^ (1 mM) did not support the binding (Figure 2b). Inhibitory anti-αvβ3 mAb LM609 weakly but significantly suppressed the binding of soluble αvβ3 to CCL5 in 1mM Mn^2+^ (Figure 2c), indicating that the binding of CCL5 to αvβ3 is specific. Previous studies showed that the disintegrin domain of ADAM15 (ADAM15 disintegrin) specifically bound to integrin αvβ3 [37] and later αIIbβ3 [39]. We found that ADAM15 disintegrin suppressed the binding of soluble αvβ3 to the immobilized CCL5, but GST did not (Figure 2d), indicating that CCL5 specifically bound to αvβ3. These findings indicate that CCL5 is a new ligand for αvβ3. These findings suggest that CCL5 binds to site 1 of αvβ3 in Mn^2+^-dependent manner.

Since docking simulation predicts that CCL5 binds to site 2 of αvβ3 (docking energy −20.2) (Figure 1b), we studied whether CCL5 activates soluble αvβ3 by binding to site 2 in ELISA-type activation assays. To study whether CCL5 activates αvβ3, soluble αvβ3 was pre-incubated with CCL5 for 10 min at room temperature and then incubated with immobilized γC399tr, a specific ligand for αvβ3, in the presence of 1 mM Ca^2+^. CCL5 enhanced the binding of soluble αvβ3 to γC399tr in a dose-dependent manner (Figure 2e), but required high concentrations (>1 µg/mL) of CCL5 for detection. We found that CCL5 activated soluble αvβ3 in 1 mM Ca^2+^ in a time-dependent manner (Figure 2f).

### 3.2. CCL5 Binds to and Activates Soluble αIIbβ3

Since docking simulation predicted that CCL5 binds to site 1 and site 2 of integrin αvβ3, and bound to and activated integrin αvβ3, we studied whether CCL5 binds to and activates integrin αIIbβ3. We found that CCL5 bound to soluble αIIbβ3 in a dose-dependent manner (Figure 3a). However, heat treatment (80 °C 10 min) or specific inhibitors for αIIbβ3, anti-β3 mAb 7E3, or eptifibatide (0.6 µg/mL), did not affect the binding of soluble αIIbβ3 to CCL5 (Appendix A). However, ADAM15 disintegrin fused to GST, a specific ligand to αIIbβ3, suppressed the binding of soluble αIIbβ3 to the immobilized CCL5, but control GST did not (Figure 3b), indicating that CCL5 specifically bound to αIIbβ3. These findings indicate that CCL is a new ligand for αIIbβ3.

Docking simulation predicts that CCL5 binds to the allosteric site (site 2) of αvβ3 at high affinity. We previously showed that peptides from site 2 bind to CX3CL1, CXCL12, and several integrin activators (e.g., sPLA2-IIA and CD40L), and blocked allosteric activation integrins αvβ3, α4β1 or α5β1 [13,14,40], indicating that they actually bind to site 2 and binding to site 2 is required for allosteric activation. We found that cyclic site 2 peptides from β3 and β1 bound to CCL5, indicating that CCL5 actually binds to site 2 (Figure 3c). We thus studied if CCL5 activates soluble integrin αIIbβ3. 

We previously developed ELISA-type activation assays, in which soluble integrins are incubated with ligands that are immobilized to plastic in the presence of chemokines or other activators in 1 mM Ca^2+^ (to keep integrins inactive) in cell-free conditions [13]. Integrin activation is defined as the increase in binding of soluble integrins to immobilized ligand. To measure the increase in ligand binding affinity, we need to use monovalent ligand. Pac-1 IgM specific for αIIbβ3, multivalent ligand-mimetic antibody with potential 10-binding sites, has been widely used for detecting αIIbβ3 ligand binding. Since monovalent Fab of Pac-1 IgM (kindly provided by Sandy Shattil, UC San Diego, La Jolla, CA, USA) showed very weak affinity to αIIbβ3, we generated the C-terminal residues 390-411 of fibrinogen γ-chain fused to GST (designated γC390-411) as a ligand for αIIbβ3. αIIbβ3 binds to the C-terminal 400HHLGGAKQAGDV411 sequence of fibrinogen γ-chain C-terminal domain (γC) [41]. We showed that soluble recombinant αIIbβ3 (activated by 1 mM Mn^2+^) bound to γC390-411 in ELISA-type assays in a dose dependent manner (Appendix A). Integrin αvβ3 does not recognize this sequence for binding to γC [36,42]. Using the whole fibrinogen γ-chain C-terminal domain (γC151-411) [36] as a ligand for soluble αIIbβ3 in the ELISA-type activation assay, we obtained results very similar to those with γC390-411 (Appendix A). Since γC151-411 also binds to other integrins such as αvβ3 and αMβ2 [36,43], we used γC390-411 as a ligand for αIIbβ3 throughout the entire project.

To study whether CCL5 activate αIIbβ3, soluble αIIbβ3 was pre-incubated with CCL5 for 10 min at room temperature and then incubated with immobilized γC390-411 in the presence of 1 mM Ca^2+^. CCL5 enhanced the binding of soluble αIIbβ3 to γC390-411 in a dose-dependent manner (Figure 3d). We found that cyclic site 2 peptides from β3 and β1 effectively suppressed CCL5-induced αIIbβ3 activation (Figure 3e), indicating that CCL5 is required to bind to site 2 of αIIbβ3 to activate αIIbβ3. We found that it took >30 min to fully activate soluble αIIbβ3 by CCL5 (Figure 3f). This is consistent with the previous findings that the activation of soluble integrin αvβ3 by CXCL12 was slow and required high CXCL12 concentrations [14]. These findings suggest that activation of soluble αIIbβ3 by CCL5 does not require inside-out signaling, but requires high concentrations of CCL5 and takes >60 min to complete. 

It has been assumed that 1 mM Mn^2+^ fully activates integrins [44,45,46,47]. We compared the levels of activation of soluble αIIbβ3 by CCL5 with that of 1 mM Mn^2+^ as a standard integrin activator. Unexpectedly, we found that CCL5 was much more potent (4.5-fold) than 1 mM Mn^2+^ (Figure 3g). 

### 3.3. CXCL12 Binds to and Activates Soluble αIIbβ3

CXCL12 is another chemokines that is stored in platelet granules and rapidly transported to the surface upon platelet activation [48]. We studied if CXCL12 binds to and activates integrin αIIbβ3. We obtained results similar to that of CCL5. Briefly, CXCL12 bound to αIIbβ3 in 1 mM Mn^2+^ (Figure 4a) and this interaction was suppressed by another αIIbβ3 ligand, ADAM15 (Figure 4b). CXCL12 activated αIIbβ3 in a dose-dependent manner in cell-free conditions. A high concentration of CXCL12 (>50 µg/mL) was required to fully activate soluble αIIbβ3 (Figure 4c). Previous studies showed that activation of αvβ3, α4β1 or α5β1 by CXCL12 was blocked by site 2-derived peptides [14]. Activation of αIIbβ3 by CXCL12 was also suppressed by cyclic site 2 peptides (Figure 4d), indicating that CXCL12 binding to site 2 is required for αIIbβ3 activation. It took more than 60 min to fully activate soluble αIIbβ3 by CXCL12 (Figure 4e), indicating that this activation process is relatively slow. CXCL12 was more potent (3.5-fold) in activating αIIbβ3 than that of Mn^2+^ (Figure 4f). These findings suggest that CXCL12 also play a role in αIIbβ3 activation. 

### 3.4. CX3CL1 Binds to and Activates Soluble αIIbβ3

Transmembrane CX3CL1 is not stored in platelet granules but is expressed on the surface of activated endothelial cells [6]. Previous studies showed that platelet–endothelial interaction plays a role in vascular inflammation (e.g., atherosclerosis) [25]. We hypothesized that CX3CL1 on endothelial cells mediates platelet–endothelial interaction by binding to and activating αIIbβ3. We obtained results with the chemokine domain of CX3CL1 similar to that of CCL5. Briefly, we found that CX3CL1 bound to αIIbβ3 in 1 mM Mn^2+^ (Figure 5a) and this interaction was suppressed by another αIIbβ3 ligand, ADAM15 (Figure 5b). CX3CL1 activated soluble αIIbβ3 in ELISA-type activation assays in 1 mM Ca^2+^ and high concentrations of CXCL1 (>50 µg/mL) was required to fully activate αIIbβ3 (Figure 5c). This activation was suppressed by cyclic site 2 peptides (Figure 5d), indicating that CX3CL1 binding to site 2 is required for αIIbβ3 activation. It took more than 60 min to fully activate soluble αIIbβ3 by CX3CL1 (Figure 5e), indicating that this activation process is relatively slow. CX3CL1 was several times more potent in activating αIIbβ3 than that of Mn^2+^ (Figure 5f). These findings suggest that CX3CL1 on activated endothelial cells mediates platelet–endothelial interaction by activating and binding to platelet αIIbβ3. Moreover, αIIbβ3 can be activated by soluble CX3CL1 in circulation. 

### 3.5. CCL5, CXCL12, and CX3CL1 more Efficiently Activate Cell-Surface αIIbβ3 Than Soluble αIIbβ3

αIIbβ3 on CHO cells is not activated by platelet agonists, since intracellular machineries for inside-out activation are missing [49]. Previous studies have shown that integrin activation by CX3CL1 or CXCL12 in CHO cells is independent of their receptors (CX3CR1 and CXCR4, respectively) [13,14]. CHO cells do not express CCR3 [50]. Additionally, response to CCL5 in CHO cells required transfection of CCR1 or CCR5 [51,52]. Therefore, it is highly likely that activation of integrin αIIbβ3 by CCL5, CXCL12, and CX3CL1 in CHO cells occurs independent of their cognate receptors. We studied whether CCL5, CXCL12, and CX3CL1 activate cell-surface αIIbβ3 on CHO cells using FITC-labeled γC390-411 in 1 mM Ca^2+^ in flow cytometry. We found that CCL5, CXCL12, or CX3CL1 markedly enhanced the binding of γC390-411 to cell-surface αIIbβ3 (Figure 6a). Notably, the levels of activation by CCL5, CXCL12, and CX3CL1 was much higher than 1 mM Mn2+ (Figure 6b), which is consistent with the results in the ELISA-type activation assays. We studied the time-course of activation of cell-surface αIIbβ3 by CCL5, CXCL12, and CX3CL1. CCL5, CXCL12, and CX3CL1 activated cell-surface αIIbβ3 on CHO cells much more rapidly (half-maximal response < 1 min) (Figure 6c) than in soluble αIIbβ3. Activation of soluble αIIbβ3 by CCL5, CXCL12, or CX3CL1 was slow and required more than 60 min to reach maximum).

We studied the activation of cell-surface αIIbβ3 as a function of CCL5, CXCL12, and CX3CL1 concentrations. We found that activation of cell-surface αIIbβ3 was detectable at 1−10 ng/mL concentrations (Figure 7a−c). We suspect that this is probably because soluble chemokines are rapidly concentrated on the cell surface by binding to cell-surface proteoglycans [14]. These findings suggest that activation of platelet surface αIIbβ3 can be efficiently activated by these chemokines in an allosteric manner in the absence of canonical inside-out signaling by these chemokines, and this process is biologically potentially important.

## 4. Discussion

The present study establishes that CCL5 bound to site 1 of αvβ3, and activated αvβ3 by binding to site 2. This indicates that CCL5 is similar to CX3CL1 and CXCL12 in binding to and activating integrin αvβ3 [13,14]. Next, we showed that chemokines CCL5, CXCL12, and CX3CL1 bound to platelet integrin αIIbβ3, indicating that they are new ligands for αIIbβ3 (Figure 8a). Furthermore, activation of soluble integrin αIIbβ3 was suppressed by cyclic site 2 peptides, suggesting that the binding of these chemokines to site 2 is required. Since this activation is observed in cell-free conditions, inside-out signaling or their cognate receptors is not involved in this activation. High concentrations (>10 µg/mL) of the chemokines were required to detect activation of soluble αIIbβ3 and this activation was a slow process. These chemokines activated cell-surface αIIbβ3 on CHO cells, which lack machinery for inside-out signaling and cognate receptors for chemokines. In contrast to activation of soluble αIIbβ3, activation of cell-surface αIIbβ3 can be detected at very low concentrations of chemokines (1−10 ng/mL) and much more quickly (half maximal response <1 min) than that of soluble αIIbβ3, probably because soluble chemokines can be quickly concentrated on the cell surface by binding to cell-surface proteoglycans. These findings are the first evidence that platelet integrin αIIbβ3 can be efficiently activated by chemokines independent of canonical inside-out signaling. We propose that this is one of the mechanisms for enhancing ligand binding affinity of αIIbβ3, which is distinct from the canonical inside-out signaling (Figure 8b). CCL5 and CXCL12, which are stored in platelet granules, and CX3CL1 on the surface of activated endothelial cells (Figure 8c), and those in circulation (e.g., during inflammation and cytokine storm) (Figure 8b) may possibly contribute to αIIbβ3 activation. We propose that chemokine binding to site 2 of αIIbβ3 is a potential target for drug discovery and site 2 peptides have potential as therapeutics for thrombosis.

Since activation of integrins by binding of chemokines to site 2 (allosteric activation) enhanced the binding of monovalent ligand to integrins, we propose that site 2-mediated allosteric integrin activation enhanced ligand affinity of αIIbβ3. It is possible that inside-out signaling induces integrin binding to multivalent ligands (e.g., integrin clustering) and site 2-mediated allosteric binding enhances ligand affinity. Since transport of platelet granules and their contents to the surface is known to require inside-out signaling, allosteric activation by chemokines also requires inside-out signaling. It is imperative to determine the role of each pathway in the future studies.

Binding of integrins to their ligands requires the presence of divalent cations, and different cations can markedly alter integrin affinities to fibronectin [44], and Mn^2+^ produced the most striking increase in integrin ligand affinity compared with other divalent cations (Mg^2+^ or Ca^2+^) in a wide variety of integrins. Subsequently, Mn^2+^ has been widely used as a positive control for integrin activation. Mn^2+^-induced integrin activation was thought to mimic physiologic integrin activation because Mn^2+^ activates integrins in the absence of a bound ligand and induces similar epitope exposure [53]. It is surprising that CX3CL1, CXCL12, and CCL5 are more effective in activating αIIbβ3 than 1 mM Mn^2+^. It is likely that αIIbβ3 is less sensitive to Mn^2+^ than other integrins. It is possible to use chemokines as the standard of αIIbβ3 activation in future studies.

It is unclear if αIIbβ3 activation mediated by the binding of chemokines site 2 requires global conformational changes in αIIbβ3. Previous studies have shown that sPLA2-IIA allosterically activated integrins αvβ3, α4β1, and α5β1 [37]. sPLA2-IIA did not change reactivity to activation-dependent anti-β1 antibodies HUTS4 and HUTS21 [54,55]. In our preliminary experiments, chemokines did not change reactivity of conformation-dependent anti-β3 integrin antibody LIBS2 [56] in 1 mM Ca^2+^ (Appendix A). It is thus possible that allosteric activation of αIIbβ3 by chemokines may not induce global conformational changes as in allosteric activation of β1 integrins [40]. It has recently been reported that the binding of allosteric activating anti β1 mAb TS2/16 to integrin α6β1 did not change the 3D structure of the integrin at all [57]. It would be important to study whether the binding of chemokines to site 2 induces changes in 3D structure of integrins in future experiments.

## Figures and Tables

**Figure 1 cells-11-03059-f001:**
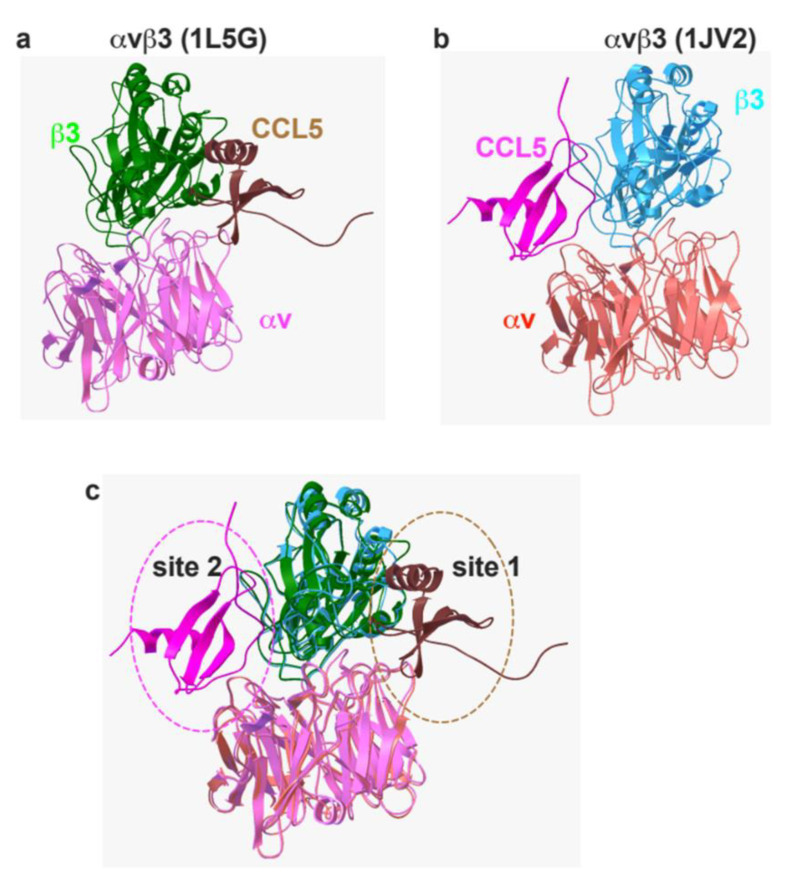
Model of the interaction between CCL5 and integrin αvβ3. Docking simulation between CCL5 (1EQT.pdb) and integrin αvβ3 (**a**) with open-headpiece conformation 1L5G.pdb or (**b**) with closed headpiece conformation, 1JV2.pdb using Autodock3, as described in the Methods section; the docking models were superimposed (**c**). Different colors indicate specific protein chain.

**Figure 2 cells-11-03059-f002:**
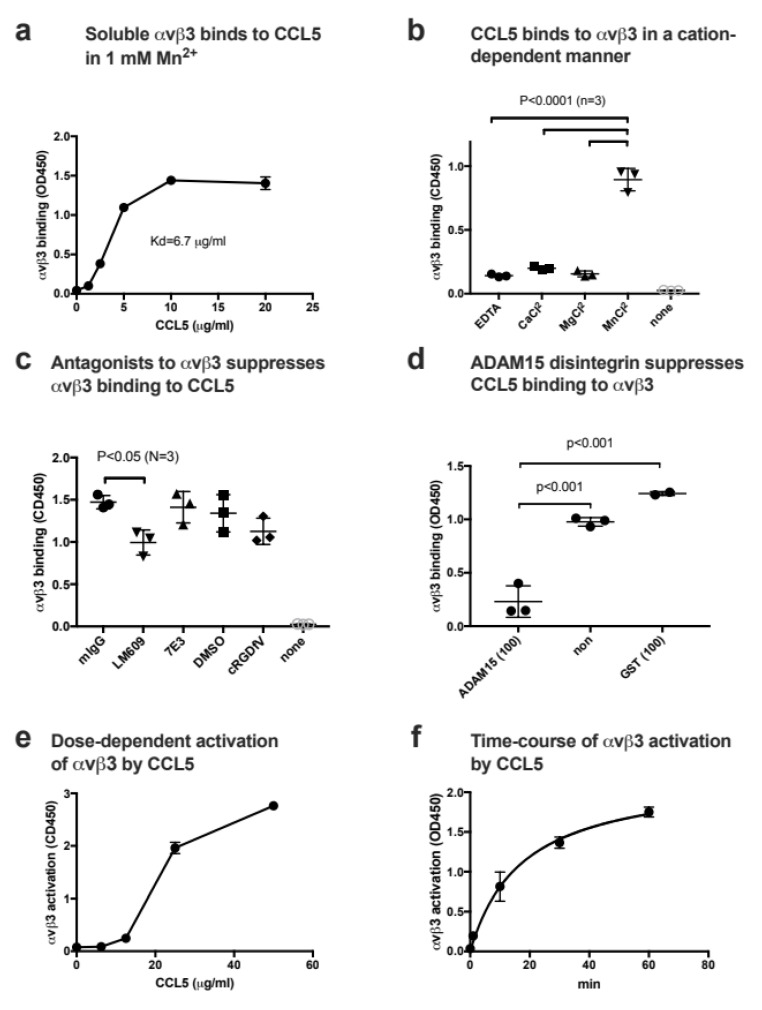
CCL5 binds to and activates integrin αvβ3. (**a**) Binding of soluble integrin αvβ3 to CCL5 in 1 mM Mn^2+^ in cell-free conditions. Wells of 96-well microtiter plates were coated with CCL5 and remaining protein-binding sites were blocked with BSA. Wells were incubated with soluble αvβ3 (1 µg/mL) in Tyrode-HEPES buffer with 1 mM Mn^2+^ (to activate αvβ3) for 1 h at room temperature. After washing the wells to remove unbound integrin, bound αvβ3 was quantified using anti-β3 mAb (AV10) and HRP-conjugated anti mouse IgG. Data are shown as means +/− SD in triplicate experiments. (**b**) Cation dependency of αvβ3 binding to CCL5. Binding of soluble integrin αvβ3 to CCL5 was measured in the presence of different cations (1 mM) in ELISA-type binding assays. Data are shown as means +/− SD in triplicate experiments. (**c**) Effect of antagonists to αvβ3 on CCL5 binding to αvβ3. The concentrations of antagonists used are 10 µg/mL for LM609 and 7E3, and 10 µM for cRGDfV. Data are shown as means +/− SD in triplicate experiments. (**d**) ADAM15 disintegrin, a specific ligand for αvβ3, suppresses CCL5 binding to αvβ3. ADAM15 disintegrin fused to GST or control GST (100 µg/mL each) were included in the binding assay as described in (**a**). Data are shown as means +/− SD in triplicate experiments. (**e**) Activation of soluble αvβ3 by CCL5 in 1 mM Ca^2+^. Wells of 96-well microtiter plates were coated with γC399tr, a specific αvβ3-ligand (20 µg/mL) and remaining protein-binding sites were blocked with BSA. Wells were incubated with soluble αvβ3 (1 µg/mL) in Tyrode-HEPES buffer with 1 mM Ca^2+^ for 1 h at room temperature in the presence of CCL5. After removing the unbound integrin, bound αvβ3 was quantified using anti-β3 mAb (AV10) and HRP-conjugated anti mouse IgG. Data are shown as means +/− SD in triplicate experiments. (**f**) Time-course of activation of soluble αvβ3 by CCL5. Wells of 96-well microtiter plates were coated with γC399tr (a specific ligand for αvβ3) (20 µg/mL) and remaining protein-binding sites were blocked with BSA. Wells were incubated with soluble αvβ3 (1 µg/mL) and CCL5 (20 µg/mL) at room temperature in Tyrode-HEPES buffer with 1 mM Ca^2+^ for the time indicated. After washing the wells to remove unbound integrin, bound αvβ3 was quantified using anti-β3 mAb (AV10) and HRP-conjugated anti mouse IgG. Data are shown as means +/− SD in triplicate experiments.

**Figure 3 cells-11-03059-f003:**
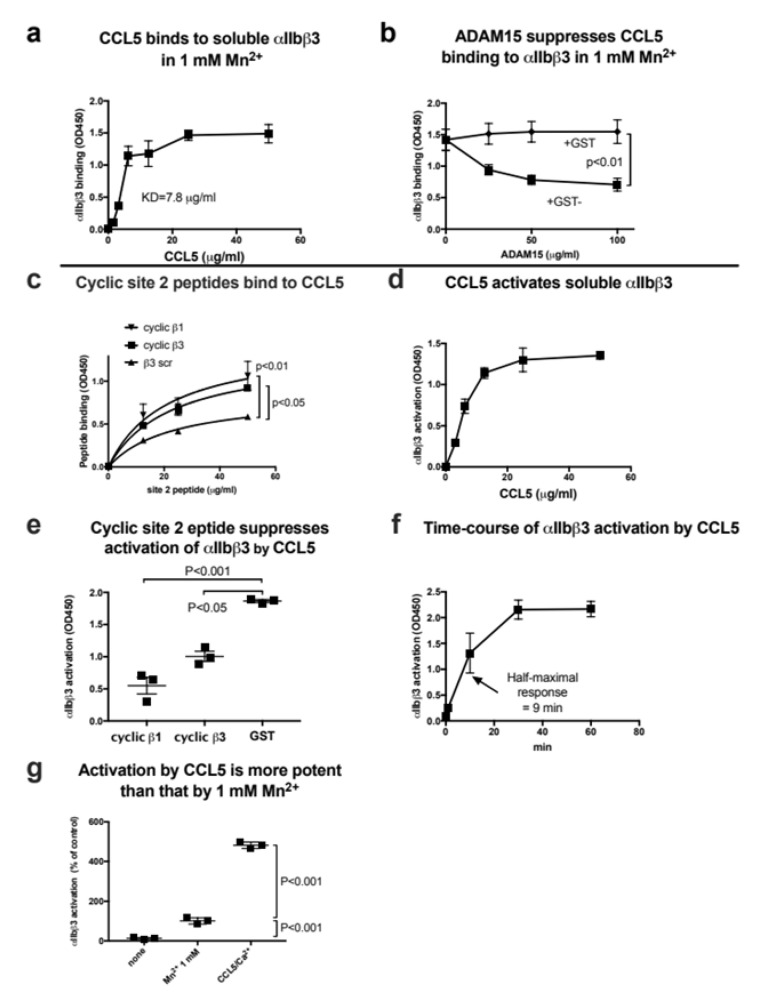
CCL5 binds to and activates soluble integrin αIIbβ3 in cell-free conditions. (**a**) Binding of soluble integrin αIIbβ3 to CCL5 in 1 mM Mn^2+^ in cell-free conditions. Wells of 96-well microtiter plates were coated with CCL5 and remaining protein-binding sites were blocked with BSA. Wells were incubated with soluble αIIbβ3 (1 µg/mL) in Tyrode-HEPES buffer with 1 mM Mn^2+^ (to activate αIIbβ3) for 1 h at room temperature. After washing the wells to remove unbound integrin, bound αIIbβ3 was quantified using anti-β3 mAb (AV10) and HRP-conjugated anti mouse IgG. Data are shown as means +/− SD in triplicate experiments. (**b**) ADAM15 disintegrin, another ligand for αIIbβ3, suppressed CCL5 binding to αIIbβ3. ADAM15 disintegrin fused to GST or control GST were included in the binding assay as described in (**a**). (**c**) Binding of cyclic site 2 peptides to CCL5. Wells of 96-well microtiter plates were coated with CCL5 (20 µg/mL) and remaining protein binding sites were coated with BSA. Wells were then incubated with cyclic β3 or β1 site 2 peptide fused to GST or control β3 scrambled peptide for 1 h at room temperature and bound site 2 peptide was quantified using HRP-conjugated anti-GST antibodies. Data are shown as means +/− SD in triplicate experiments. (**d**) Activation of soluble αIIbβ3 by CCL5 in 1 mM Ca^2+^. Wells of 96-well microtiter plates were coated with γC390-411 (the αIIbβ3-ligand peptide conjugated to GST) (20 µg/mL) and the remaining protein-binding site were blocked with BSA. Wells were incubated with soluble αIIbβ3 (1 µg/mL) in Tyrode-HEPES buffer with 1 mM Ca^2+^ for 1 h at room temperature in the presence of CCL5. After removing the unbound integrin, bound αIIbβ3 was quantified using anti-β3 mAb (AV10) and HRP-conjugated anti mouse IgG. Data are shown as means +/− SD in triplicate experiments. (**e**) Effect of site 2 peptide on αIIbβ3 activation by CCL5. Activation of αIIbβ3 by CCL5 was assayed as described in (d) except that cyclic site 2 peptides or control GST (100 µg/mL each) were included as a competitor. Data are shown as means +/− SD in triplicate experiments. (**f**) Time-course of activation of soluble αIIbβ3 by CCL5. Wells of 96-well microtiter plates were coated with γC390-411 (20 µg/mL) and remaining protein-binding sites were blocked with BSA. Wells were incubated with soluble αIIbβ3 (1 µg/mL) and CCL5 (20 µg/mL) at room temperature in Tyrode-HEPES buffer with 1 mM Ca^2+^ for the time indicated. After washing the wells to remove unbound integrin, bound αIIbβ3 was quantified using anti-β3 mAb (AV10) and HRP-conjugated anti mouse IgG. Data are shown as means +/− SD in triplicate experiments. (**g**) Comparison of αIIbβ3 activation by CCL5 and that by 1 mM Mn^2+^. Soluble αIIbβ3 was activated with only 1 mM Mn^2+^ or CCL5 (50 µg/mL) in 1 mM Ca^2+^. Data are shown as means +/− SD in triplicate experiments.

**Figure 4 cells-11-03059-f004:**
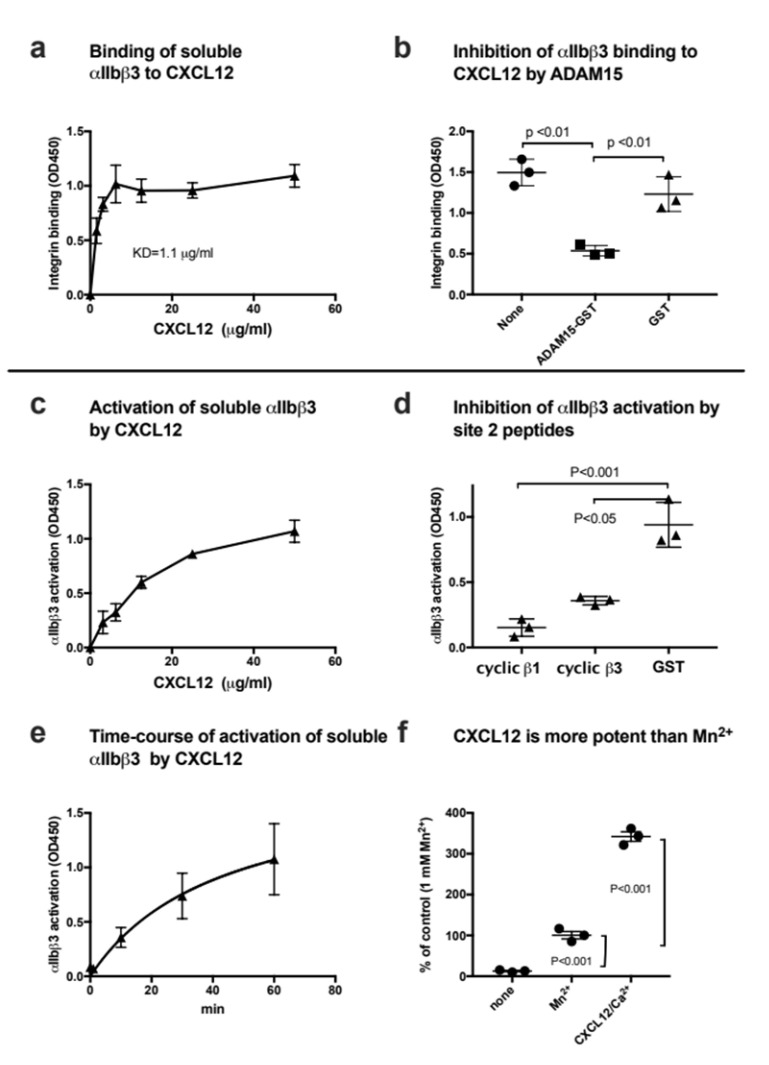
CXCL12 binds to and activates soluble integrin αIIbβ3 in cell-free conditions. (**a**) Binding of soluble integrin αIIbβ3 to CXCL12 in 1 mM Mn^2+^ in cell-free conditions. Wells of 96-well microtiter plates were coated with CXCL12 and remaining protein-binding sites were blocked with BSA. Wells were incubated with soluble αIIbβ3 (1 µg/mL) in Tyrode-HEPES buffer with 1 mM Mn^2+^ (to activate αIIbβ3) for 1 h at room temperature. After washing the wells to remove unbound integrin, bound αIIbβ3 was quantified using anti-β3 mAb (AV10) and HRP-conjugated anti mouse IgG. Data are shown as means +/− SD in triplicate experiments. (**b**) ADAM15 disintegrin, another ligand for αIIbβ3, suppressed CXCL12 binding to αIIbβ3. ADAM15 disintegrin fused to GST or control GST (100 µg/mL each) were included in the binding assay as described in (a). Data are shown as means +/− SD in triplicate experiments. (**c**) Activation of soluble αIIbβ3 by CXCL12 in 1 mM Ca^2+^. Wells of 96-well microtiter plates were coated with γC390-411 (20 µg/mL) and remaining protein-binding sites were blocked with BSA. Wells were incubated with soluble αIIbβ3 (1 µg/mL) in Tyrode-HEPES buffer with 1 mM Ca^2+^ for 1 h at room temperature in the presence of CXCL12. After removing the unbound integrin, bound αIIbβ3 was quantified using anti-β3 mAb (AV10) and HRP-conjugated anti mouse IgG. Data are shown as means +/− SD in triplicate experiments. (**d**) Effect of site 2 peptide on αIIbβ3 activation by CXCL12. Activation of αIIbβ3 by CXCL12 was assayed as described in (**c**) except that cyclic site 2 peptides fused to GST or control GST (100 µg/mL) were included as a competitor. Data are shown as means +/− SD in triplicate experiments. (**e**) Time-course of activation of soluble αIIbβ3 by CXCL12. Wells of 96-well microtiter plates were coated with γC390-411 (20 µg/mL) and remaining protein-binding sites were blocked with BSA. Wells were incubated with soluble αIIbβ3 (1 µg/mL) and CXCL12 (20 µg/mL) at room temperature in Tyrode-HEPES buffer with 1 mM Ca^2+^ for the time indicated. After washing the wells to remove unbound integrin, bound αIIbβ3 was quantified using anti-β3 mAb (AV10) and HRP-conjugated anti mouse IgG. Data are shown as means +/− SD in triplicate experiments. (**f**) Comparison of αIIbβ3 activation by CXCL12 and that by 1 mM Mn^2+^. Soluble αIIbβ3 was activated with only 1 mM Mn^2+^ or CXCL12 (50 µg/mL) in 1 mM Ca^2+^. Data are shown as means +/− SD in triplicate experiments.

**Figure 5 cells-11-03059-f005:**
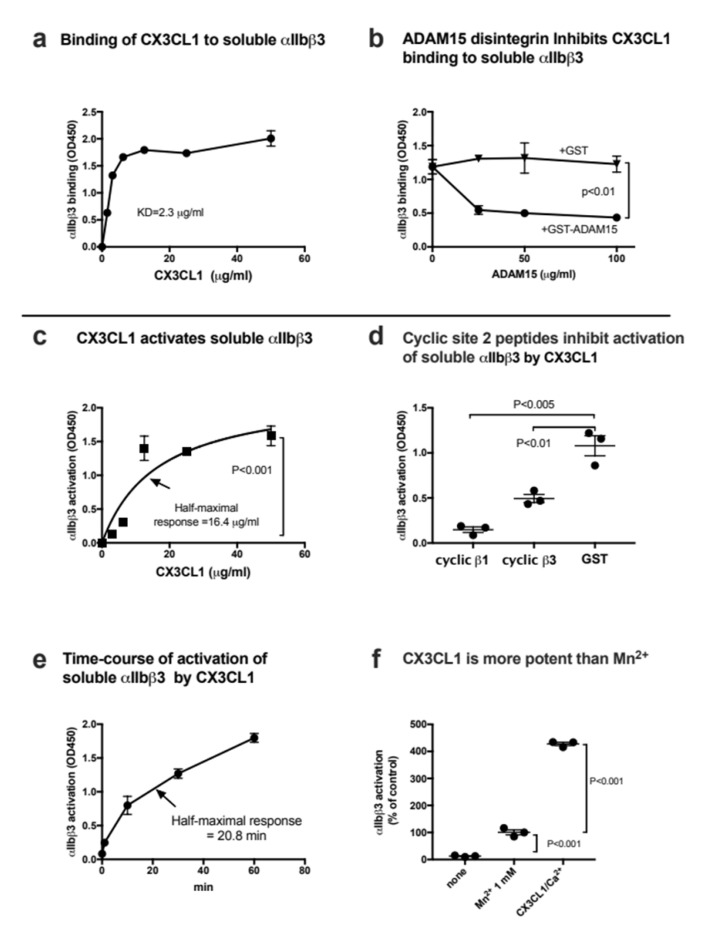
CX3CL1 binds to and activates soluble integrin αIIbβ3 in cell-free conditions. (**a**) Binding of soluble integrin αIIbβ3 to CX3CL1 in 1 mM Mn^2+^ in cell-free conditions. Wells of 96-well microtiter plates were coated with CX3CL1 and remaining protein-binding sites were blocked with BSA. Wells were incubated with soluble αIIbβ3 (1 µg/mL) in Tyrode-HEPES buffer with 1 mM Mn^2+^ (to activate αIIbβ3) for 1 h at room temperature. After washing the wells to remove unbound integrin, bound αIIbβ3 was quantified using anti-β3 mAb (AV10) and HRP-conjugated anti mouse IgG. Data are shown as means +/− SD in triplicate experiments. (**b**) ADAM15 disintegrin suppressed CX3CL1 binding to αIIbβ3. ADAM15 disintegrin fused to GST or control GST were included in the binding assay as described in (**a**). (**c**) Activation of soluble αIIbβ3 by CX3CL1 in 1 mM Ca^2+^. Wells of 96-well microtiter plates were coated with γC390-411 (20 µg/mL) and the remaining protein-binding site were blocked with BSA. Wells were incubated with soluble αIIbβ3 (1 µg/mL) in Tyrode-HEPES buffer with 1 mM Ca^2+^ for 1 h at room temperature in the presence of CX3CL1. After removing the unbound integrin, bound αIIbβ3 was quantified using anti-β3 mAb (AV10) and HRP-conjugated anti mouse IgG. Data are shown as means +/− SD in triplicate experiments. (**d**) Effect of site 2 peptide on αIIbβ3 activation by CX3CL1. Activation of αIIbβ3 by CXCL12 was assayed as described in (c) except that cyclic site 2 peptides (100 µg/mL) were included as a competitor. Data are shown as means +/− SD in triplicate experiments. (**e**) Time-course of activation of soluble αIIbβ3 by CX3CL1. Wells of 96-well microtiter plates were coated with γC390-411 (20 µg/mL) and the remaining protein-binding site were blocked with BSA. Wells were incubated with soluble αIIbβ3 (1 µg/mL) and CX3CL1 (20 µg/mL) at room temperature in Tyrode-HEPES buffer with 1 mM Ca^2+^ for the time indicated. After washing the wells to remove unbound integrin, bound αIIbβ3 was quantified using anti-β3 mAb (AV10) and HRP-conjugated anti mouse IgG. Data are shown as means +/− SD in triplicate experiments. (**f**) Comparison of αIIbβ3 activation by CX3CL1 and that by 1 mM Mn^2+^. Soluble αIIbβ3 was activated with only 1 mM Mn^2+^ or CX3CL1 (50 µg/mL) in 1 mM Ca^2+^. Data are shown as means +/− SD in triplicate experiments.

**Figure 6 cells-11-03059-f006:**
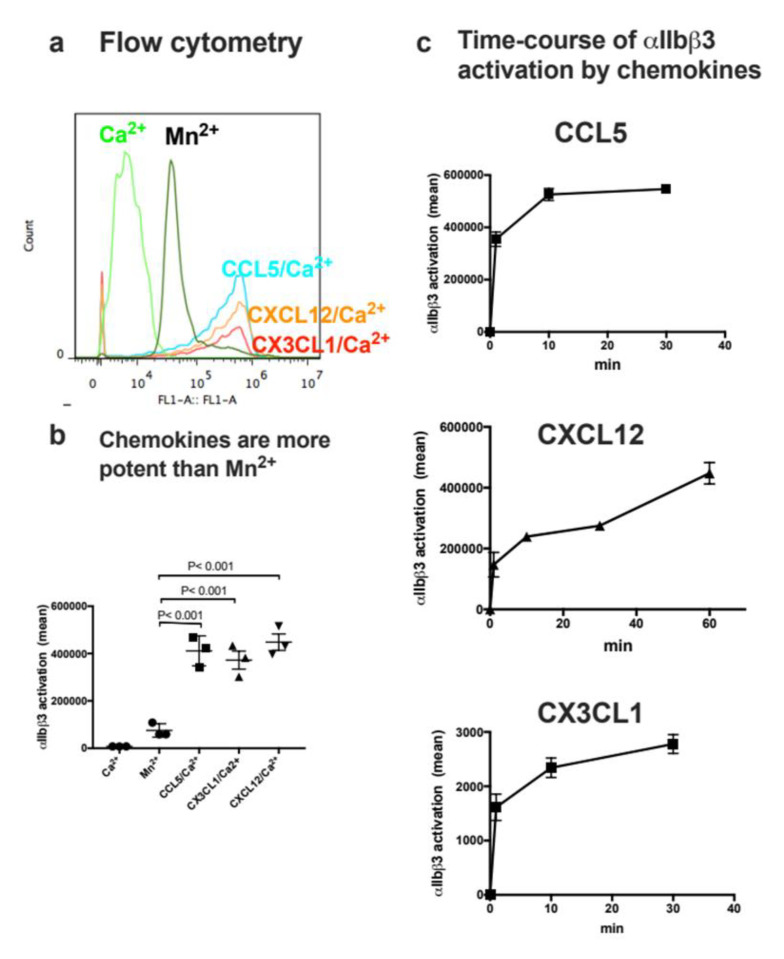
Chemokines activate cell-surface αIIbβ3 in 1 mM Ca^2+^. (**a**) Activation of cell-surface αIIbβ3 on CHO cells. αIIbβ3-CHO cells were incubated with chemokines (50 µg/mL) for 30 min on ice and then incubated with FITC-labeled γC390-411 for 30 min at room temperature. The cells were washed with PBS/0.02% BSA and analyzed by flow cytometer. Data are shown as mean fluorescence intensity +/− SD in triplicate experiments. (**b**) Comparison of activation of cell-surface αIIbβ3 by 1 mM Mn^2+^ and chemokines in 1 mM Ca^2+^. αIIbβ3-CHO cells were incubated with chemokines as described in (**a**). Data are shown as mean fluorescence intensity +/− SD in triplicate experiments. Ca^2+^ only and Mn^2+^ only are significantly different in unpaired t test (n = 3), *p* < 0.001. (**c**) Time-course of activation of cell-surface αIIbβ3 on αIIbβ3-CHO cells. αIIbβ3-CHO cells were incubated with chemokines (50 µg/mL) and FITC-labeled γC390-411 and incubated for the time indicted at room temperature. The cells were washed with PBS/0.02% BSA and analyzed by flow cytometer. Data are shown as mean fluorescence intensity +/− SD in triplicate experiments.

**Figure 7 cells-11-03059-f007:**
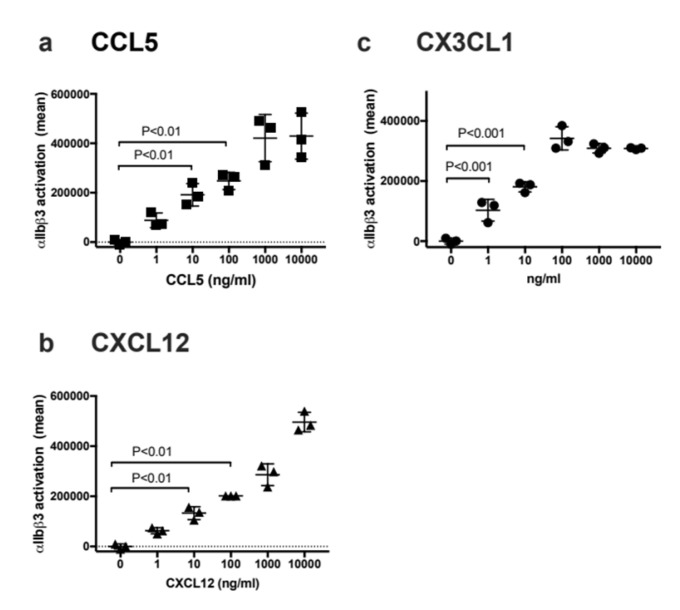
Dose-response of chemokine-induced activation of cell-surface αIIbβ3. αIIbβ3-CHO cells were incubated with chemokines (**a**) CCL5, (**b**) CXCL12, or (**c**) CX3CL1 at indicated concentrations for 30 min on ice and then incubated with FITC-labeled γC390–411 for 30 min at room temperature. The cells were washed with PBS/0.02% BSA and analyzed by flow cytometer. Data are shown as mean fluorescence intensity +/− SD in triplicate experiments.

**Figure 8 cells-11-03059-f008:**
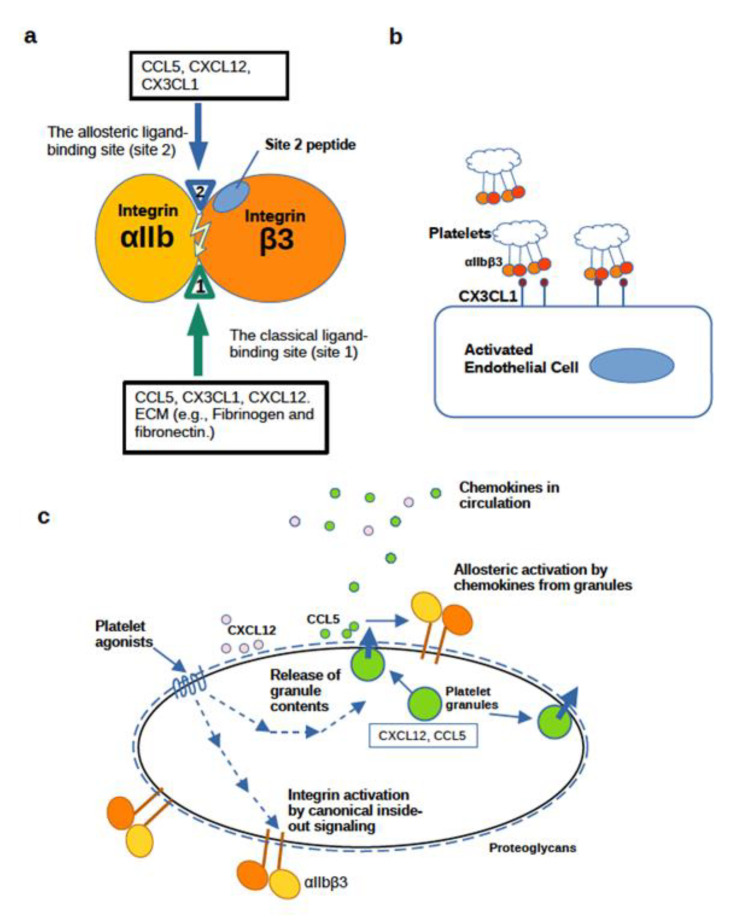
Possible biological role of allosteric activation of αIIbβ3 by CCL5, CXCL12, and CX3CL1. (**a**) A model of the binding of CCL5, CXCL12, and CX3CL1 to the classical ligand binding site (site 1) and to the allosteric site (site 2) in integrin αIIbβ3. CCL5, CXCL12, and CX3CL1 bind to site 1 and activate integrins by binding to site 2. The site 2 peptide inhibits integrin activation by chemokines. (**b**) Activation of αIIbβ3 is induced by CXCL12 or CCL5 stored in platelet granules upon platelet activation or circulating CCL5, CXCL12, or CX3CL1 overproduced during inflammation. Several chemokines are stored in platelet granules and are transported to the platelet surface upon platelet activation by platelet agonists (e.g., thrombin, collagen, and ADP). Platelet integrin αIIbβ3 is inactive in resting platelets but quickly activated upon platelet activation. This activation is mediated by canonical inside-out signaling. In the present study we showed that the ligand-binding affinity of αIIbβ3 is enhanced by the binding of chemokines to site 2. Note that transfer of granular contents to the surface is mediated by inside-out signaling by platelet agonists. Levels of pro-inflammatory chemokines in circulation increase during inflammation and they may also allosterically activate αIIbβ3. This may be a missing link between inflammation and thrombosis. (**c**) Potential role of CX3CL1 in platelet–endothelial cell interaction. Platelet–endothelial cell interaction is involved in vascular inflammation (e.g., atherosclerosis). Transmembrane CX3CL1 is expressed on endothelial cells activated by pro-inflammatory cytokines (e.g., IL-1 β and TNFα). CX3CL1 on the cell surface can activate αIIbβ3 by binding to site 2 and then support platelet–endothelial interaction by binding to site 1.

**Table 1 cells-11-03059-t001:** Amino acid residues involved in CCL5 (1EQT.pdb) and αvβ3 (1L5G.pdb, open headpiece) predicted by docking simulation.

CCL5	αv	β3
Phe12, Ala13, Tyr14, Ile15, Ala16, Arg17, Pro18, Met19, Pro20, Arg21, Ala22, His23, Thr43, Lys45, Arg47, Val49, Cys50, Asn52, Glu54, Lys55, Lys56, Trp57, Arg59, Glu60 Tyr61	Met118, Lys119, Gln145, Asp146, Ile147, Asp148, Asp150, Gly151, Tyr178, Thr212, Ala215, Ile216, Phe217, Asp218, Asp219, Arg248,	Asp119, Ser121. Tyr122, Ser123, Met124, Lys125, Asp126, Asp127, Tyr166, Asp179, Met180, Thr182, Arg214, Asn215, Arg216, Asp217, Ala218, Pro219, Glu220, Asp251, Ala252, Lys253, Asn313,

Amino acid residues within 0.6 nm between CCL5 and αvβ3 were selected using Swiss PDB Viewer (version 4.1) (Swiss Institute of Bioinformatics, Basel, Swiss).

**Table 2 cells-11-03059-t002:** Amino acid residues involved in CCL5 and αvβ3 (1JV2.pdb, closed headpiece) predicted by docking simulation.

CCL5	αv	β3
Thr7, Thr8, Cys10, Phe12, Ala13, Tyr14, Ile15, Ala16, Arg17, Pro18, Met19, Pro20, Arg21, His23, Ser35, Asn36, Pro37, Thr43, Lys45, Arg47, Gln48, Val49, Asn52, Glu55, Lys56, Trp57, Tyr61	Glu15, Gly16, Tyr18, Lys42, Asn44, Thr45, Thr46, Gln47, Pro48, Gly49, Ile50, Val51, Glu52, Ser90, His91	Lys159, Pro160, Val161, Ser162, Met165, Glu171, Glu174, Asn175, Pro186, Met187, Phe188, **Val266, Gln267, Val275, Gly276, Ser277, Asp278, His280, Tyr281, Ser282, Ala283, Thr285, Thr286**

Amino acid residues within 0.6 nm between CCL5 and αvβ3 were selected using Pdb Viewer (version 4.1). Amino acid residues in β3 that are in the cyclic site 2 peptide are in bold.

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
