# Peer review of "Pro-Inflammatory Chemokines CCL5, CXCL12, and CX3CL1 Bind to and Activate Platelet Integrin αIIbβ3 in an Allosteric Manner"

_cells, 2022, doi:10.3390/cells11193059_

Round 1
Reviewer 1 Report
This manuscript does an excellent job demonstrating αIIbβ3 could be activated allosterically. Overproduction of cytokines during inflammation may activate this receptor, which could be the missing link between inflammation and thrombosis.
Despite the docking study, it should make an addition molecular dynamic caculation to compare energy stability site1 vs site2 etc. MD study will get more specific details.
Introduction
p. 1, lines 42: The format of "-" after 1β is strange please keep the same format
Results
p. 3, line 111: format of αvβ3 is not the same of "α4β1" and "α5β1"
p. 4, line 127: description of different colors indicate specific protein chain should be added in the Figure 1 subtitles
References:
p. 17, line 586,589 and 626: minor format issue should be revised.
Author Response
This manuscript does an excellent job demonstrating αIIbβ3 could be activated allosterically. Overproduction of cytokines during inflammation may activate this receptor, which could be the missing link between inflammation and thrombosis.
Despite the docking study, it should make an addition molecular dynamic calculation to compare energy stability site1 vs site2 etc. MD study will get more specific details.
Response: We are not ready for MD study and it is not possible to include it this time.
Introduction
p. 1, lines 42: The format of "-" after 1β is strange please keep the same format
Response: corrected.
Results
p. 3, line 111: format of αvβ3 is not the same of "α4β1" and "α5β1"
Response: corrected.
p. 4, line 127: description of different colors indicate specific protein chain should be added in the Figure 1 subtitles
Response: the statement was included
References:
p. 17, line 586,589 and 626: minor format issue should be revised.
Response: corrected.
Reviewer 2 Report
In this manuscript Takada and co-workers propose three pro-inflammatory chemokines, CCL5, CXCL12 and CX3CL1 as novel ligands of alphaIIbbeta3, the most abundant, relevant and widely investigated platelet integrin. The study is mainly biochemical, based on in vitro binding and activation studies complemented by activation in the established model of CHO cells expressing alphaIIbbeta3. This report follows previous studies by the Takada laboratory and as such the experiments are sound and well executed. The findings have clear implications for the regulation of integrins at the crossroads of inflammation and haemostasis. Some functional assays using platelets to demonstrate the functional relevance of those findings would have made up for a round paper; without these the study, albeit important, remains preliminary. Nevertheless, I recommend publication of this study; however there are a number of minor issues that would need revision:
1. The abstract abruptly starts with an outline of results. What is the justification of the study? What is the hypothesis?
2. In the introduction, lines 81-82, remove this sentence, it states the obvious but does not provide justification. I’m however missing a couple of sentences that clearly link previous work on CXCL12 and CX3CL1 to the purpose of the present study (or the hypothesis) before introducing CCL5 as a protein stored in platelet granules. In the last paragraph of the introduction, condense by removing unnecessary experimental detail and speculation (like in lines 95-96). Lines 101-102, “might contribute” and “might be a possible” rather than “contribute” and “is a possible”. The statements are not supported by functional data and remain speculative.
3. I don’t seem to see a description of the docking procedure in the Methods section, please provide.
4. Is it standard performing binding essays at room temperature rather than 37°C?
5. Figure 2a, provide the Kd for CCL5. This is done for the other ligands in figures 3a and 4a. I also recommend mentioning those values in the text. Importantly, is there any reason why the concentrations of these three ligands are expressed in ug/ml rather than in molar terms? This does not make out for meaningful comparisons between these three ligands. Please clarify.
6. There are several instances of “data not shown” throughout the text, particularly on pg 6. These data should be provided, eventually as supplemental information. They appear to me as important to show.
7. Figures 6, 7 and 8 should perhaps be arranged by assay rather than by chemokine. In other words, panels a and b of the three figures as one separate figure, panels c as another figure and panels d as a third figure. This would better match the way the panels are described in the text. Alternatively make a single, large figure by merging figures 6, 7, 8. Panels a and be should be described together: panel a is a representative flow cytometry plot and panel b plots the flow cytometry data from 3 experiments. Consequently, the sentence in lines 373-375 should come before line 369, so that panel b is mentioned before panel c.
8. In the discussion, make more explicit reference to panels a, b and c of figure 9. Perhaps move some of the information in the legend (especially of panel c) to the main text. On line 468 “prerequisite” is probably not the best choice of word. You mean “imperative” or something similar? On line 478, what do you mean with “it is possible to use chemokines…”? It certainly is possible, are you suggesting or recommending that? If so, state it clearly.
Other corrections:
-In Cells the methods section comes after the introduction, you may be asked to change that.
-The usage of the English language is good; there are however a number of grammar errors. I recommend a careful round of proof reading.
-Line 15 “We show that inflammatory…” rather than “We showed that…”
-Line 33, extracellular matrix (ECM) proteins,
-Line 55, introduce the SDF-1 abbreviation (and spell it out) the first time CXCL12 is presented
-Line 56, Heterotrimeric G protein…
-Line 67, provide a reference for this statement. What do you mean with “we have used…”?
-Line 88, a transplantation is not a disorder, do the authors mean transplant rejection or something related?
-Line 223, “but requires…” to the end of the sentence. This is not an interpretation, just repetition of the results. Modify or delete.
-Line 373, specify what you mean with “see above”. Where exactly?
-Line 382, “and this process is biologically relevant”. This statement is not supported by functional data at this stage, please tone down.
Author Response
In this manuscript Takada and co-workers propose three pro-inflammatory chemokines, CCL5, CXCL12 and CX3CL1 as novel ligands of alphaIIbbeta3, the most abundant, relevant and widely investigated platelet integrin. The study is mainly biochemical, based on in vitro binding and activation studies complemented by activation in the established model of CHO cells expressing alphaIIbbeta3. This report follows previous studies by the Takada laboratory and as such the experiments are sound and well executed. The findings have clear implications for the regulation of integrins at the crossroads of inflammation and haemostasis. Some functional assays using platelets to demonstrate the functional relevance of those findings would have made up for a round paper; without these the study, albeit important, remains preliminary. Nevertheless, I recommend publication of this study; however there are a number of minor issues that would need revision:
1. The abstract abruptly starts with an outline of results. What is the justification of the study? What is the hypothesis?
Response: Abstract was revised and included justification and hypothesis. Thank you.
2. In the introduction, lines 81-82, remove this sentence, it states the obvious but does not provide justification. I’m however missing a couple of sentences that clearly link previous work on CXCL12 and CX3CL1 to the purpose of the present study (or the hypothesis) before introducing CCL5 as a protein stored in platelet granules. In the last paragraph of the introduction, condense by removing unnecessary experimental detail and speculation (like in lines 95-96). Lines 101-102, “might contribute” and “might be a possible” rather than “contribute” and “is a possible”. The statements are not supported by functional data and remain speculative.
Response: Introduction was revised.
3. I don’t seem to see a description of the docking procedure in the Methods section, please provide.
Response: the docking procedure was included.
4. Is it standard performing binding essays at room temperature rather than 37°C?
Response: It is standard to do binding assay at room temperature in my lab.
5. Figure 2a, provide the Kd for CCL5. This is done for the other ligands in figures 3a and 4a. I also recommend mentioning those values in the text. Importantly, is there any reason why the concentrations of these three ligands are expressed in ug/ml rather than in molar terms? This does not make out for meaningful comparisons between these three ligands. Please clarify.
Response: KD for CCL5 was included. Chemokines we use have similar in size, so we usually use μg/ml. We will include molecular weight in the manuscript to make it possible to calculate molar terms.
6. There are several instances of “data not shown” throughout the text, particularly on pg 6. These data should be provided, eventually as supplemental information. They appear to me as important to show.
Response: We included the data as a supplemental file.
7. Figures 6, 7 and 8 should perhaps be arranged by assay rather than by chemokine. In other words, panels a and b of the three figures as one separate figure, panels c as another figure and panels d as a third figure. This would better match the way the panels are described in the text. Alternatively make a single, large figure by merging figures 6, 7, 8. Panels a and be should be described together: panel a is a representative flow cytometry plot and panel b plots the flow cytometry data from 3 experiments. Consequently, the sentence in lines 373-375 should come before line 369, so that panel b is mentioned before panel c.
Response: Figs 6-8 was reorganized as new Fig, 6 and 7.
8. In the discussion, make more explicit reference to panels a, b and c of figure 9. Perhaps move some of the information in the legend (especially of panel c) to the main text. On line 468 “prerequisite” is probably not the best choice of word. You mean “imperative” or something similar? On line 478, what do you mean with “it is possible to use chemokines…”? It certainly is possible, are you suggesting or recommending that? If so, state it clearly.
Response: Fig. 9 (new Fig. 8 a-c) was mentioned in the discussion. The text was revised.
Other corrections:
-In Cells the methods section comes after the introduction, you may be asked to change that.
Response: The position of the method section was changed.
-The usage of the English language is good; there are however a number of grammar errors. I recommend a careful round of proof reading.
-Line 15 “We show that inflammatory…” rather than “We showed that…”
-Line 33, extracellular matrix (ECM) proteins,
-Line 55, introduce the SDF-1 abbreviation (and spell it out) the first time CXCL12 is presented
-Line 56, Heterotrimeric G protein…
-Line 67, provide a reference for this statement. What do you mean with “we have used…”?
-Line 88, a transplantation is not a disorder, do the authors mean transplant rejection or something related?
-Line 223, “but requires…” to the end of the sentence. This is not an interpretation, just repetition of the results. Modify or delete.
-Line 373, specify what you mean with “see above”. Where exactly?
-Line 382, “and this process is biologically relevant”. This statement is not supported by functional data at this stage, please tone down.
Response: reference added. Revised the text.